# EXTENSION OF PHYSICS-INFORMED NEURAL NETWORKS TO SOLVING PARAMETERIZED PDEs

**Woojin Cho**[1,3], **Minju Jo**[1,2], **Haksoo Lim**[1],
**Kookjin Lee**[3], **Dongeun Lee**[4], **Sanghyun Hong**[5], **Noseong Park**[6]
[1]Yonsei University, [2]LG CNS, [3]Arizona State University
[4]Texas A&M University-Commerce, [5]Oregon State University, [6]KAIST
{snowmoon, alflsowl12, limhaksoo96}@yonsei.ac.kr
kookjin.lee@asu.edu
eundong4@gmail.com
sanghyun.hong@oregonstate.edu
noseong@kaist.ac.kr

## ABSTRACT

In this paper, we address PINNs' problem of repetitive and time-consuming training by proposing a novel extension, parameterized physics-informed neural networks ($P^2$INNs). $P^2$INNs enable modeling the solutions of parameterized PDEs via explicitly encoding a latent representation of PDE parameters. With the extensive empirical evaluation, we demonstrate that $P^2$INNs outperform the baselines both in accuracy and parameter efficiency on benchmark 1D and 2D parameterized PDEs and are also effective in overcoming the known "failure modes".

## 1 INTRODUCTION

Scientific machine learning (SML) (Baker et al., 2019) has evolved rapidly, requiring exact satisfaction of important physical laws. Among various deep-learning models that encode such physical characteristics (Raissi et al., 2019; Lee & Carlberg, 2021; Cranmer et al., 2020b; Lee et al., 2021; Satorras et al., 2021), physics-informed neural networks (PINNs) (Raissi et al., 2019) are gaining traction in the research community. This is primarily due to their sound computational formalism, which enforces governing physical laws to learn solutions.

PINNs parameterize the solution $u(x,t)$ of partial differential equations (PDEs) using a neural network $u_\Theta(x,t)$ that takes the spatial and temporal coordinates $(x,t)$ as input and has $\Theta$ as the model parameters. During training, the neural network minimizes a *PDE residual loss* (cf. equation 11) denoting the governing equation, at a set of collocation points, and a *data matching loss* (cf. equation 10 and equation 12), which enforce initial/boundary conditions, at another set of collocation points sampled from initial/boundary conditions. This computational formalism enables to infuse the physical laws, described by the governing equation $\mathcal{F}(x,t,u)$, into the solution model and, thus, is denoted as "physics informed". PINNs have shown to be effective in solving many different PDEs, such as Navier–Stokes equations (Shukla et al., 2021; Jagtap & Karniadakis, 2020; Jagtap et al., 2020). While powerful, PINNs suffer from several obvious weaknesses.

**W1**) PDE operators are highly nonlinear (making training extremely difficult);
**W2**) Repetitive trainings from scratch are needed when solutions to new PDEs are sought (even for new PDEs arising from new PDE parameters in parameterized PDEs).

There have been various efforts to mitigate each of these issues: (For addressing W1) curriculum-learning-type training algorithms that train PINNs from easy PDEs to hard PDEs[1] (Krishnapriyan et al., 2021), and (for addressing W2) meta-learning PINNs (Liu et al., 2022); or directly learning solutions of parameterized PDEs such that $u_\Theta(x,t;\boldsymbol{\mu})$, where $\boldsymbol{\mu}$ is a set of PDE parameters. However, there has been a less focus on addressing both problems in a unified PINN framework.

---

[1]Following the notational conventions in curriculum learning, we use the terms, "easy" and "hard," to indicate data that are easy or hard for neural networks to learn.

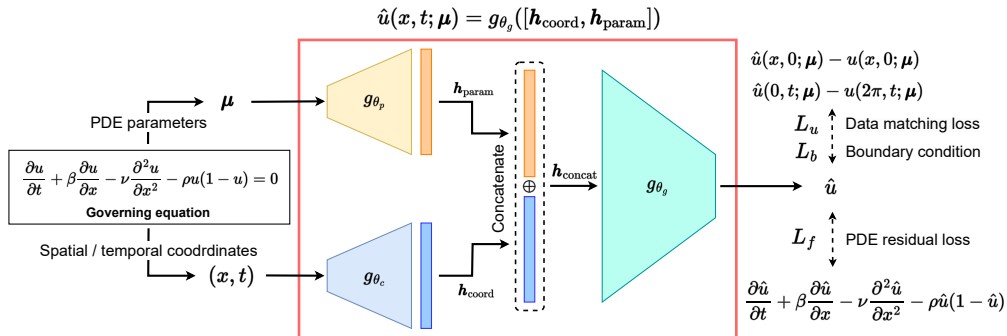

Figure 1: **P²INNs architecture.** The first two encoders $g_{\theta_p}$ and $g_{\theta_c}$ are added to generate better representations for the PDE parameter and the spatial/temporal coordinate. We also carefully customize the manifold network $g_{\theta_g}$. In this figure, we provide the CDR equation as an example.

To mitigate the both issues in W1 and W2 simultaneously, we propose a variant of PINNs for solving parameterized PDEs, called *parameterized physics-informed neural networks* (P²INNs). P²INNs approximate solutions as a neural network of a form $u_\Theta(x, t; \boldsymbol{\mu})$ (for resolving W2) and show reasonable accuracy (cf. Table 1) even for harder PDEs (for resolving W1). A novel modification in our approach is to explicitly extract a hidden representation of the PDE parameters by employing a separate encoder network, $\boldsymbol{h}_{\text{param}} = g_{\Theta_p}(\boldsymbol{\mu})$, and uses this hidden representation to parameterize the solution neural network, $u_\Theta(x, t; \boldsymbol{h}_{\text{param}})$. Rather than simply treating $\boldsymbol{\mu}$ as a coordinate in the parameter domain, we infer useful information of PDEs from the PDE parameters $\boldsymbol{\mu}$, constructing the latent manifold on which the hidden representation of each PDE lies.

To demonstrate the effectiveness of the proposed model, we demonstrate the performance of the proposed model with well-known benchmarks (Krishnapriyan et al., 2021), i.e., parameterized CDR equations. As studied in Krishnapriyan et al. (2021), certain choices of the PDE parameters (e.g., high convective or reaction term) make training PINNs very challenging (i.e., harder PDEs), and our goal is to show that the proposed method is capable of producing approximate solutions with reasonable accuracy for those harder PDEs.

## 2 PROPOSED METHODS

Now we introduce our parameterized physics-informed neural networks (P²INNs). In essence, our goal is to design a neural network architecture that effectively emulates the action of the parameterized PDE solution function, $u(x, t; \boldsymbol{\mu})$.

### 2.1 MODEL ARCHITECTURE

For P²INNs, we propose a modularized design of the neural network $u_\Theta(x, t; \boldsymbol{\mu})$, which consists of three parts, i.e., two separate encoders $g_{\theta_p}$ and $g_{\theta_c}$, and a manifold network $g_{\theta_g}$ such that

$$u_\Theta(x, t; \boldsymbol{\mu}) = g_{\theta_g}([g_{\theta_c}(x, t); g_{\theta_p}(\boldsymbol{\mu})]), \tag{1}$$

where $\Theta = \{\theta_c, \theta_p, \theta_g\}$ denotes the set of model parameters (cf. Figure 1). The two encoders, $g_{\theta_c}$ and $g_{\theta_p}$, take the spatiotemporal coordinate $(x, t)$ and the PDE parameters $\boldsymbol{\mu}$ as inputs and extract hidden representations such that $\boldsymbol{h}_{\text{coord}} = g_{\theta_c}(x, t)$ and $\boldsymbol{h}_{\text{param}} = g_{\theta_p}(\boldsymbol{\mu})$. The two extracted hidden representations are then concatenated and fed into the manifold network to infer the solution of of the PDE with the parameters $\boldsymbol{\mu}$ at the coordinate $(x, t)$, i.e., $\hat{u}(x, t; \boldsymbol{\mu}) = g_{\theta_g}([\boldsymbol{h}_{\text{coord}}; \boldsymbol{h}_{\text{param}}])$.

The important design choice here is that we explicitly encode the PDE parameters into a hidden representation as opposed to treating the PDE parameters merely as a coordinate in the parameter domain, e.g., $(x, t, \boldsymbol{\mu})$ is combined and directly fed into our ablation model, called PINN-P, for our ablation study in Appendix. With the abuse of notation, P²INNs can be expressed as a function of $(x, t)$, parameterized by the hidden representation: $u_\Theta(x, t; \boldsymbol{\mu}) = u_{\{\theta_c, \theta_g\}}(x, t; \boldsymbol{h}_{\text{param}})$. This expression emphasizes our intention that we explicitly utilize the PDE model parameters to characterize the behavior of the solution neural network.

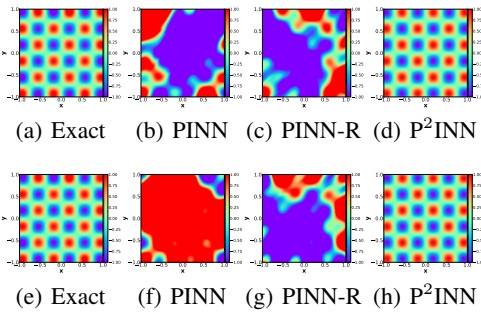

(a) Exact    (b) PINN    (c) PINN-R    (d) P$^2$INN

(e) Exact    (f) PINN    (g) PINN-R    (h) P$^2$INN

Figure 2: [2D Helmholtz equation] Exact solutions and results of baselines and P$^2$INN for $a$=2.7 (seen) (a-d) and $a$=2.75 (unseen) (e-h).

Table 1: **P$^2$INNs greatly improve the quality of CDR solutions.** We also compare the average $L_2$ absolute (Abs.) and relative (Rel.) errors of PINN and P$^2$INNs in six different CDR equations. IMP is the improvement ratio of P$^2$INNs to PINN.

| PDE type | PINN | | P$^2$INN | | IMP. (%) | |
|---|---|---|---|---|---|---|
| | Abs. | Rel. | Abs. | Rel. | Abs. | Rel. |
| **Convection** | 0.0496 | 0.0871 | **0.0159** | **0.0295** | 67.93 | 66.12 |
| **Diffusion** | 0.3611 | 0.6939 | **0.1878** | **0.3655** | 47.98 | 47.33 |
| **Reaction** | 0.5825 | 0.6431 | **0.0029** | **0.0049** | 99.50 | 99.24 |
| **Conv.-Diff.** | 0.1493 | 0.2793 | **0.0601** | **0.1177** | 59.74 | 57.86 |
| **Reac.-Diff.** | 0.4744 | 0.5614 | **0.1529** | **0.2306** | 67.78 | 58.92 |
| **Conv.-Diff.-Reac.** | 0.3231 | 0.3847 | **0.0360** | **0.0583** | 88.87 | 84.85 |

**Encoder for equation input**    The equation encoder $g_{\theta_p}$ reads the PDE parameters, and generates a hidden representation of the equation, denoted as $\boldsymbol{h}_{\text{param}}$. We employ the following fully-connected (FC) structure for the encoder:

$$\boldsymbol{h}_{\text{param}} = \sigma(FC_{D_p} \cdots (\sigma(FC_2(\sigma(FC_1(\boldsymbol{\mu})))))), \tag{2}$$

where $\sigma$ denotes a non-linear activation, such as ReLU and tanh, and $FC_i$ denotes the $i$-th FC layer of the encoder. $D_p$ means the number of FC layers of $g_{\theta_p}$. We note that $\boldsymbol{h}_{\text{param}}$ has a size larger than that of $\boldsymbol{\mu}$ in our design to encode the space and time-dependent characteristics of the parametrized PDE. Since highly non-linear PDEs show different characteristics at different spatial and temporal coordinates, we intentionally employ relatively high-dimensional encoding.

**Encoder for spatial and temporal coordinate input**    The spatial and temporal coordinate encoder $g_{\theta_c}$ generates a hidden representation $\boldsymbol{h}_{\text{coord}}$ for $(x, t)$. This encoder has the following structure:

$$\boldsymbol{h}_{\text{coord}} = \sigma(FC_{D_c} \cdots (\sigma(FC_2(\sigma(FC_1(x, t)))))), \tag{3}$$

where $FC_i$ and $D_c$ denote the $i$-th FC layer and the number of FC layers of $g_{\theta_c}$, respectively.

**Manifold network**    The manifold network $g_{\theta_g}$ reads the two hidden representations, $\boldsymbol{h}_{\text{param}}$ and $\boldsymbol{h}_{\text{coord}}$, and infer the input equation's solution at $(x, t)$, denoted as $\hat{u}(x, t; \boldsymbol{\mu})$. With the inferred solution $\hat{u}$, we construct two losses, $L_u$ and $L_f$. The manifold network can have various forms but we use the following form:

$$\hat{u}(x, t; \boldsymbol{\mu}) = \sigma(FC_{D_g} \cdots \sigma(FC_1(\boldsymbol{h}_{\text{concat}}))), \text{ where } \boldsymbol{h}_{\text{concat}} = \boldsymbol{h}_{\text{coord}} \oplus \boldsymbol{h}_{\text{param}}. \tag{4}$$

Here, $\oplus$ represents the concatenation of the two vectors; $D_g$ denotes the number of FC layers of $g_{\theta_g}$.

## 2.2 TRAINING

Model training is performed by minimizing the regular PINN loss. With the prediction $\hat{u}$ produced by P$^2$INNs, our basic loss function consists of three terms as follows:

$$L(\Theta) = w_1 L_u + w_2 L_f + w_3 L_b,, \tag{5}$$

where $L_u$, $L_b$, and $L_f$ enforces initial, boundary conditions, and physical laws in PDEs, respectively, and $w_1, w_2, w_3 \in \mathbb{R}$ are hyperparameters. In general, the overall training method follows the training procedure of the original PINN (Raissi et al., 2019). The only exception is that the PDE residual loss associated with multiple PDEs is minimized in a mini-batch whereas in the original PINN, the residual of only one PDE is minimized. To be more specific, in each iteration, we create a mini-batch of $\{\boldsymbol{\mu}_i, (x_i, t_i)\}_{i=1}^{B}$, where $B$ is a mini-batch size. We randomly sample the collocation points and, thus, there can be multiple different PDEs, identified by $\boldsymbol{\mu}_i$, in a single mini-batch.

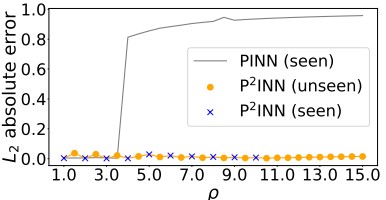

Figure 3: [Reaction equation] Interpolation and extrapolation results for unseen $\rho$.

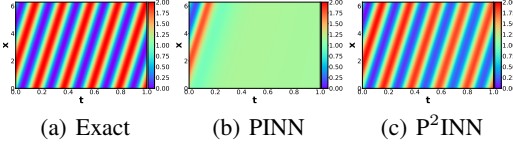

(a) Exact      (b) PINN      (c) P²INN

Figure 4: Failure modes in the convection equation of $\beta = 30$ (a-c). P²INNs much more accurately predict reference solutions.

## 3   EVALUATION

In this section, we test the performance of P²INNs on the benchmark PDE problems, 1D CDR equations and 2D Helmholtz equations (cf. Appendix A), both of which are known to suffer from the failure modes. For 1D CDR equations, we employ 6 different equation types stemmed from CDR equations with the varying parameters as listed in Table 2. The experimental results are summarized in Table 1 and Figure 2. To evaluate the performance of the model, we measure the $L_2$ relative and absolute errors between the solution predicted by the model and the analytic solution. Due to space reasons, we put detailed experimental setups and results are in Appendix.

**Methodology.** We train PINN and PINN-R for each parameter configuration in each equation type — in other words, there are as many models as the number of PDE parameter configurations for an equation type. To train P²INNs, however, we train it with all the collocation points of the multiple parameter configurations in each equation type, following the training method in Section 2.2. Therefore, we have only one model in each equation type. See Appendix for detailed discussion.

**Experimental results.** The values for 6 CDR equations in Table 1 represent the averaged errors for the coefficient ranges of 1 to 5, 1 to 10, and 1 to 20 (cf. Table 3). In Table 1, whereas PINNs show fluctuating performance, our P²INNs show stable performance for all the equation types. The most notable improvement ratios are observed for the reaction and the convection-diffusion-reaction equations. For instance, PINN marks an absolute error of 0.5825 whereas P²INN achieves an error of 0.0029 for the reaction equations, i.e., 200 time smaller error. Since large coefficients incur equations difficult to solve, PINNs commonly fail in the range.

For 2D Helmholtz equations, we train models with $a \in [2.5, 3.0]$ with interval 0.1, and then test them with interval 0.05. Notably, as shown in Figure 2, P²INNs consistently shows good performance in both cases where $a$ is a seen parameter ($a = 2.7$) and an unseen ($a = 2.75$) parameter. However, PINN and PINN-R struggle, despite the fact that both of these values are within the seen(trained) parameter range for two models. This reaffirms the robustness and efficacy of our proposed P²INN approach in higher-dimensional settings, where there are non-trivial boundary conditions.

**Inferring Solutions of Unseen PDE Parameters** We further evaluate our P²INNs in more challenging situations: testing trained models on PDE parameters that are unseen during training, which can be considered as *real-time multi-query* scenarios. For reaction equations, we train P²INNs on $\rho \in [1, 10]$ with interval 1 and conduct interpolation on $\rho \in [1.5, 9.5]$ with interval 1 and extrapolation on $\rho \in [10.5, 15]$ with interval 0.5. As shown in Figure 3, PINNs' failure for $\rho > 4$ contrasts P²INNs' exceptional performance, demonstrating its resilience in extrapolation, not limited good performance only for learned or closely aligned parameters.

**P²INNs in PINN's Failure Modes** It is well known that PINNs have several failure cases. Among the reported failure cases of PINN, one notable case is the convection equation with $\beta = 30$. In our experimental settings, $\beta = 30$ corresponds to an extrapolation task after being trained for up to $\beta = 20$, which is considered as one of the most challenging task. These two equations generate signals sharply fluctuating over time. As shown in Figure 4 and Table 4, therefore, PINNs fail to predict reference solutions whereas our method almost exactly reproduce them (cf. Appendix I).

## 4    RELATED WORK

In the recent literature, PINNs have evolved in many different ways to resolve issues inherent with the vanilla PINNs. Some architectural enhancements have been made in Cho et al. (2024b) (a low-rank extension PINNs for model efficiency and a hypernetwork for effective training) and in Cho et al. (2024a) (a separable design of model parameters for efficient training). A systematic assessment for PINNs and a new sample strategie have been investigated in PINNACLE Lau et al. (2023). There have been some effort to combine PINNs with symbolic regression in universal PINNs Podina et al. (2023) and to devise a preconditioner for PINNs from an PDE operator preconditioning perspective De Ryck et al. (2023). Lastly, novel optimizers for effective training of PINNs have been proposed in Yao et al. (2023) (MultiAdam) and Müller & Zeinhofer (2023) (based on natural gradient descent).

## 5    CONCLUSION

PINN is a highly applicable and promising technology for many engineering and scientific domains. However, due to the highly nonlinear characteristic of PDEs, PINNs show very poor performance in certain parameterized PDE problems. In addition, there is a weakness that the model must be re-trained from scratch to analyze a new PDE. To solve these chronic issues, we propose parameterized physics-informed neural networks ($P^2$INNs), which can learn similar parameterized PDEs simultaneously. Through this approach, it is possible to overcome the failure situation of PINNs that could not be solved in previous studies. To ensure that it is effective in general cases, we use more than thousands of CDR equations and show that $P^2$INNs outperform baselines in almost all cases of the benchmark PDEs.

## ACKNOWLEDGMENTS

This work was supported by the Institute of Information & Communications Technology Planning & Evaluation (IITP) grant funded by the Korean government (MSIT) (No. 2020-0-01361, Artificial Intelligence Graduate School Program ay Yonsei University)

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

## A    DATASETS

For simplicity but without loss of generality, we assume the parameterized 1D CDR equations and 2D Helmholtz equations (cf. equation 6 and equation 7). To generate the ground-truth data, we use either analytic or numerical solutions. In case of 1D CDR equations, we analyze the target equations with three types of initial conditions $u(x,0)$: two Gaussian distributions of $N(\pi, (\pi/2)^2)$ and $N(\pi, (\pi/4)^2)$, and a sinusoidal function of $1 + \sin(x)$. To solve the equation, we use the Strang splitting method (Strang, 1968). For 2D Helmholtz equations, we obtain the exact solution by calculating it directly.

### A.1    1D CONVECTION-DIFFUSION-REACTION EQUATIONS

We consider parameterized CDR equations:

$$\frac{\partial u}{\partial t} + \beta \frac{\partial u}{\partial x} - \nu \frac{\partial^2 u}{\partial x^2} - \rho u(1-u) = 0, \ x \in \Omega, \ t \in [0, T]. \tag{6}$$

The equation describes how the state variable $u$ changes over time with the existence of convective (the second term), diffusive (the third term), and reactive (the fourth term) phenomena. Here, $\beta$ is a coefficient about how fast transportable the equation is, $\nu$ is a diffusivity for the diffusion phase, and $\rho$ is a scaling parameter about spreading velocity. Note that we choose the well-known Fisher's form $\rho u(1 - u)$, which was used in Krishnapriyan et al. (2021), as our reaction term.

Each of these individual PDEs has their own importance and has been studied extensively:

1. Convection-diffusion equations are used in fluid dynamics, particle chemistry, computational finance, and so on,
2. Reaction-diffusion equations are popular in the domain of biophysics and mathematical biology,
3. Convection equations, diffusion equations, and reaction equations are for describing transport, diffusive, and reactive phenomena, respectively in simplified settings.

In total, there are six classes of Convection-Diffusion-Reaction equations, each of which has its own importance in science. For each dataset, we select 1,000 collocation points, 256 initial points, 100 boundary points, and 1,000 test points.

### A.2    2D HELMHOLTZ EQUATIONS

$$\frac{\partial^2 u(x,y)}{\partial x^2} + \frac{\partial^2 u(x,y)}{\partial y^2} + k^2 u(x,y) - q(x,y) = 0$$
$$q(x,y) = (-(a_1\pi)^2 - (a_2\pi)^2 + k^2)\sin(a_1\pi x)\sin(a_2\pi y) \tag{7}$$

$$u(x,y) = k^2 \sin(a_1\pi x)\sin(a_2\pi y). \tag{8}$$

We employ the specific Helmholtz equations which were used in McClenny & Braga-Neto (2020) as benchmark PDEs (cf. equation 7), and it can be directly solved as equation 8. The Helmholtz equations describe the behavior of state variable $u(x,y)$ in a 2D space, accounting for the effects of wave propagation, and a source term represented by $q(x,y)$. Here $k$ is a parameter related to wave frequency, while $a_1$ and $a_2$ control the spatial variations of the source term. In our experiments, we set $k$ to 1 and the parameters $a_1$ and $a_2$ to a common value $a$. For each dataset, we select 1,000 collocation points, 400 boundary points, and 100 test points.

## B    MOTIVATION

Our goal is to develop a method to solve parameterized PDEs via the computational formalism of PINNs' overcoming W1 and W2. With this in mind, we first attempt to obtain intuitions from the visual inspection of solution snapshots displayed on the $(x, t)$-coordinate space (Figures 5 and 6).

The first set of the examples is shown in Figure 5: The ground-truth solutions of convection equations (top row) and reaction equations (bottom row) with varying parameters $\beta$ and $\rho$, respectively.

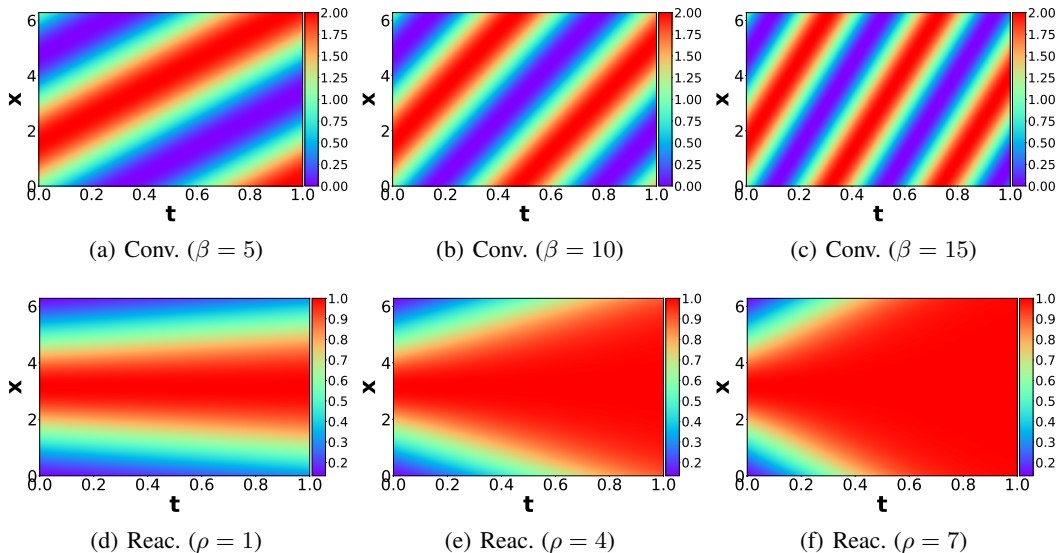

(a) Conv. ($\beta = 5$)      (b) Conv. ($\beta = 10$)      (c) Conv. ($\beta = 15$)

(d) Reac. ($\rho = 1$)      (e) Reac. ($\rho = 4$)      (f) Reac. ($\rho = 7$)

Figure 5: The ground-truth solutions of various convection equations with an initial condition of $1+\sin(x)$ (Figure 5. (a)-(c)) and reaction equations with an initial condition of a Gaussian distribution $N(\pi, (\pi/2)^2)$ (Figure 5. (d)-(f)). We note that varied solutions are made (with similar architectures) depending on changes in coefficient.

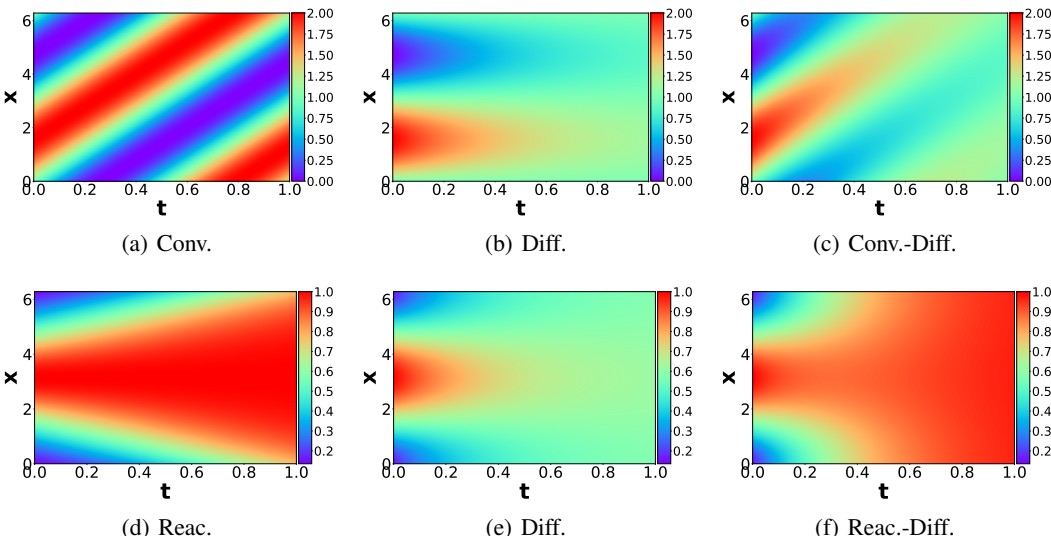

(a) Conv.      (b) Diff.      (c) Conv.-Diff.

(d) Reac.      (e) Diff.      (f) Reac.-Diff.

Figure 6: The ground-truth solutions of various CDR equations with an initial condition of $1+\sin(x)$ (Figure 6. (a)-(c)) or a Gaussian distribution $N(\pi, (\pi/2)^2)$ (Figure 6. (d)-(f)). We note that the solution in the last column reflects the first two columns' solutions. Therefore, there also exist similarities across different equation types.

As we vary the PDE parameter, e.g., increasing $\beta$, we obtain gradually changing solutions (i.e., becoming more oscillatory, as we go left from Figure 5(a) to Figure 5(c)). This suggests that model parameters of PINNs for varying PDE parameters could have similar values and this can be leveraged in the training of PINNs.

This observation indeed has been investigated in Krishnapriyan et al. (2021) to solve hard PDEs for PINNs. With a higher convective term (large $\beta$), the PDE becomes a hard problem for PINNs to solve due to the spectral bias (Rahaman et al., 2019) (i.e., the solution is highly oscillatory in time).

Thus, Krishnapriyan et al. (2021) proposed a curriculum-learning algorithm which starts to feed an easier PDE and gradually increases $\beta$ until it reaches the target value. This approach, however, drops all the intermediate model parameters obtained in the course of training. Instead, in our approach, we utilize all PDE information to train a single model for the solutions of parameterized PDEs.

The second set of examples (Figure 6, the solutions of different types of PDEs) provide a similar observation as above: even for different classes of PDEs (e.g., convection, diffusion, and convection-diffusion equations), the solutions gradually change, which can be leveraged in training PINNs.

**Motivation #1: a latent space of parameterized PDEs may exist.** Since PDEs with similar parameter settings share common characteristics, we conjecture that solutions of parameterized PDEs can be embedded onto a latent space and reconstructed by using a shared decoder network.

**Motivation #2: it will be more effective to solve similar problems simultaneously.** Considering the similarities between solutions parameterized by similar PDE parameters, we conjecture that training can be improved by attempting to solve all those similar problems together — multi-task learning approaches are also based on the same intuition (Kendall et al., 2018; Ruder, 2017).

Motivated by the observations, we develop a new approach that alleviates the two known weaknesses W1 and W2 (cf. Section 1).

## C  RELATED WORK

There is plenty of literature dealing with PDEs. We briefly introduce related work mostly used in this work.

**Machine Learning Methods for Solving Partial Differential Equations.** Traditional numerical methods such as finite element methods and finite difference methods have clear pros and cons (Patidar, 2016; Li & Bettess, 1997; Srirekha et al., 2010). The more accurate the results, the more expensive the calculation of numerically approximated formulas. It means that to earn more accurate solutions, it needs to use finer grids, which implies more cost. To alleviate those cons, researchers were attracted to machine learning approaches (Karniadakis et al., 2021; Cuomo et al., 2022). After various trials like using the Galerkin or Ritz method (Rudd & Ferrari, 2015), PINNs proposed a transformative way of using deep learning for solving general governing PDEs in a physically sound and easy-to-formulate computational formalism (Raissi et al., 2019). As elaborated above, PINNs, however, possess weaknesses which must be addressed (Krishnapriyan et al., 2021): (1) there are classes of PDEs that it is difficult for PINNs to learn (e.g., PDEs exhibiting high oscillation or sharp transitions in spatial and/or temporal domains) and (2) gradient-based training often converges to a local optimum of models. Another line of research for solving PDEs is to analyze operator learning for differential equation or deep Ritz methods (Yu et al., 2018; Li et al., 2020; Gupta et al., 2021) but PINNs still have its potential for mainly focusing on governing equations which describe physical phenomena.

**Physics as Inductive Biases.** There have been various strategies to impose physical constraints on neural networks (Cranmer et al., 2020a; Rudd & Ferrari, 2015; Lee et al., 2021). Most of them focus on imposing constraints for outputs or injecting specific physical conditions into neural networks. As a simple but effective solution, PINNs directly impose physical conditions into neural networks by using a governing equation itself as a loss (Raissi et al., 2019). This loss function is called $L_f$. In this way, PINNs can learn the residual error of the governing equation. If initial conditions are given, we can add an initial error loss term $L_u$. Furthermore, if there are specific boundary conditions, we can specify boundary conditions in $L_b$.

## D    MORE DETAILS ON EXPERIMENTAL SETUP

### D.1    LOSS

With the prediction $\hat{u}$ produced by $P^2$INNs, our basic loss function consists of three terms as follows:

$$L(\Theta) = w_1 L_u + w_2 L_f + w_3 L_b, \tag{9}$$

and $L_u$, $L_f$ and $L_b$ are defined as follows:

$$L_u = \frac{1}{N_u} \sum_{N_u} \left( \hat{u}(x, 0; \boldsymbol{\mu}) - u(x, 0; \boldsymbol{\mu}) \right)^2, \tag{10}$$

$$L_f = \frac{1}{N_f} \sum_{N_f} \left( \mathcal{F}(x, t, \hat{u}; \boldsymbol{\mu}) \right)^2, \tag{11}$$

$$L_b = \frac{1}{N_b} \sum_{N_b} \left( \hat{u}(0, t; \boldsymbol{\mu}) - \hat{u}(2\pi, t; \boldsymbol{\mu}) \right)^2, \tag{12}$$

where $N_u$, $N_f$, and $N_b$ are the cardinalities of the sets of initial conditions, collocation points, and boundary conditions; $w_1, w_2, w_3 \in \mathbb{R}$ are hyperparameters. The first and the second terms denote the data matching loss $L_u$ and the PDE residual loss $L_f$, respectively. In addition, we separately add the boundary condition term $L_b$, forcing their values equal at both top and bottom parts (see Figures 5 and 6).

### D.2    BASELINE AND ABLATION METHODS

We compare $P^2$INNs with three baselines. PINN is the original design based on fully-connected layers with non-linear activations in Raissi et al. (2019), and PINN-R is its enhancement by using residual connections, which was used in Kim et al. (2021). PINN-seq2seq (Krishnapriyan et al., 2021) is a model that applies the seq2seq learning method to the PINN model, sequentially learning data over time. We divided the entire time into 10 steps. In addition, we define one ablation model for our method, called PINN-P, which has the same structure as original PINN, but the PDE parameters $\boldsymbol{\mu}$ is treated as a coordinate in the parameter space, i.e., $(x, t, \boldsymbol{\mu})$.

Each baseline and ablation model is trained in the following way:

1. PINN, PINN-R, and PINN-seq2seq do not read PDE parameters, such as $\beta, \nu, \rho$ and $a$, but are trained separately for each of the coefficient settings.
2. PINN-P, an ablation model of $P^2$INNs, is able to process PDE parameters and is trained for all coefficient settings in each equation type.
3. Therefore, PINN, PINN-R, and PINN-seq2seq require many trained models for solving parameterized PDEs whereas PINN-P and our method require a single trained model to solve them.

**Metrics.**    The relative error and the absolute error of the $i$-th equation are defined as the averages of $\|\hat{\boldsymbol{u}}_i - \boldsymbol{u}_i\|_2 / \|\boldsymbol{u}_i\|_2$ and $\|\hat{\boldsymbol{u}}_i - \boldsymbol{u}_i\|_2$, where $i \in \{1, ..., N_e\}$ and $N_e$ is the number of equations used for the task. At this time, the errors are measured for each test points and the average value is used. We test with 3 seed numbers and report their mean.

### D.3    IMPLEMENTATION

We implement $P^2$INNs with PYTHON 3.7.11 and PYTORCH 1.10.2 that supports CUDA 11.4. We run our evaluation on a machine equipped with Intel Core-i9 CPUs and NVIDIA RTX A6000 and RTX 2080 TI GPUs.

# E MODEL CONFIGURATION AND EFFICIENCY

## E.1 DATASET STATISTICS

Table 2: Dataset statistics. For each equation type, we test three different coefficient ranges. In Conv.-Diff.-Reac., $\beta, \nu, \rho$ are non-zeros.

| Coefficient range | Convection | Diffusion | Reaction | Conv.-Diff. | Reac.-Diff. | Conv.-Diff.-Reac. |
|---|---|---|---|---|---|---|
| **1∼5** | 5 | 5 | 5 | 25 | 25 | 125 |
| **1∼10** | 10 | 10 | 10 | 100 | 100 | 1,000 |
| **1∼20** | 20 | 20 | 20 | 400 | 400 | 8,000 |

Table 2 represents dataset statistics, and our dataset generation source code is mainly based on Krishnapriyan et al. (2021).

## E.2 MODEL EFFICIENCY AND HYPERPARAMETERS

Our baselines, PINN, PINN-R, and PINN-seq2seq, are designed with 6 layers, and the dimension of hidden vector is 50. For training, we employ Adam optimizers with learning rate of $1e-3$. For our method, we set $D_p, D_c$, and $D_g$ to 4, 3, and 5 respectively. In the loss function in Eq. equation 5, we set $w_1, w_2$, and $w_3$ to 1. We use a hidden vector dimension of 50 for $g_{\theta_c}$ and $g_{\theta_g}$, and 150 for $g_{\theta_p}$. For $g_{\theta_g}$. Considering that our method is able to solve multiple equations with one model, the total model size for our method is much smaller than other baselines (see Appendix K).

# F ARCHITECTURAL DETAILS OF PINN-P

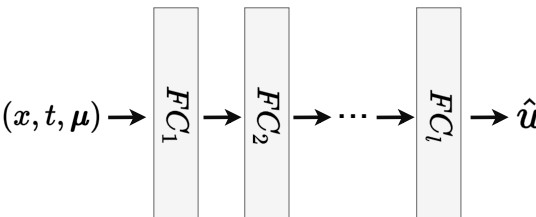

Figure 7: PINN-P architecture.

We propose PINN-P as an ablation model of our P$^2$INN. Unlike P$^2$INN, PINN-P does not have a separate encoder for PDE parameters, so that PDE parameters enter the model with coordinates. As shown in Figure 7, PINN-P consists of $l$-stacked fully-connected layers. For a fair comparison with P$^2$INNs, we set size of hidden vector to 150 and $l$ to 6, making the model size similar to P$^2$INNs.

# G REPRODUCIBILITY STATEMENT

To benefit the community, the code will be posted online. The source code for our proposed method and the dataset used in this paper are attached.

# H EXPERIMENTAL RESULTS ON 1D CDR EQUATION

Table 3: The relative and absolute $L_2$ errors over all the equations. Our P$^2$INNs surpass baselines in all but one cases, even without fine-tuning. IMP. denotes the rate of improvement of our model over the best baseline.

| | PDE type | Coefficient range | Metric | PINN | PINN-R | PINN-seq2seq | P$^2$INN | IMP. (%) |
|---|---|---|---|---|---|---|---|---|
| **Class 1** | **Convection** | 1~5 | Abs. err.
Rel. err. | 0.0183
0.0327 | 0.0222
0.0381 | 0.1281
0.2160 | **0.0039**
**0.0079** | 78.44
75.82 |
| | | 1~10 | Abs. err.
Rel. err. | 0.0164
0.0307 | 0.0666
0.1195 | 0.1924
0.3276 | **0.0093**
**0.0179** | 43.62
41.78 |
| | | 1~20 | Abs. err.
Rel. err. | 0.1140
0.1978 | 0.1624
0.2779 | 0.2252
0.3819 | **0.0198**
**0.0464** | 82.64
76.55 |
| | **Diffusion** | 1~5 | Abs. err.
Rel. err. | 0.1335
0.2733 | 0.1665
0.3462 | 0.1987
0.4050 | **0.1322**
**0.2710** | 0.97
0.84 |
| | | 1~10 | Abs. err.
Rel. err. | 0.2716
0.5259 | 0.3175
0.6206 | 0.3149
0.6174 | **0.1539**
**0.3116** | 43.34
40.75 |
| | | 1~20 | Abs. err.
Rel. err. | 0.6782
1.2825 | 0.7054
1.3401 | 0.3346
0.6442 | **0.1916**
**0.3745** | 42.74
41.87 |
| | **Reaction** | 1~5 | Abs. err.
Rel. err. | 0.3341
0.3907 | 0.3336
0.3907 | 0.4714
0.5907 | **0.0015**
**0.0027** | 99.54
99.31 |
| | | 1~10 | Abs. err.
Rel. err. | 0.6232
0.6926 | 0.3619
0.4190 | 0.6924
0.7931 | **0.0065**
**0.0089** | 98.19
97.88 |
| | | 1~20 | Abs. err.
Rel. err. | 0.7902
0.8460 | 0.4320
0.4932 | 0.8246
0.8960 | **0.0042**
**0.0092** | 99.02
98.14 |
| **Class 2** | **Conv.-Diff.** | 1~5 | Abs. err.
Rel. err. | 0.0610
0.1175 | 0.0654
0.1289 | 0.0979
0.1950 | **0.0399**
**0.0892** | 34.61
24.05 |
| | | 1~10 | Abs. err.
Rel. err. | 0.1133
0.2098 | 0.1313
0.2510 | 0.0917
0.1959 | **0.0576**
**0.1320** | 37.25
32.62 |
| | | 1~20 | Abs. err.
Rel. err. | 0.2735
0.5106 | 0.2118
0.4154 | 0.0645
0.1504 | **0.0622**
**0.1485** | 3.51
1.28 |
| | **Reac.-Diff.** | 1~5 | Abs. err.
Rel. err. | 0.1900
0.2702 | 0.1876
0.2777 | 0.4201
0.5346 | **0.1225**
**0.1856** | 34.70
31.31 |
| | | 1~10 | Abs. err.
Rel. err. | 0.5166
0.6141 | 0.3809
0.4790 | 0.6288
0.7274 | **0.1833**
**0.2756** | 51.88
42.46 |
| | | 1~20 | Abs. err.
Rel. err. | 0.7167
0.7998 | 0.7210
0.8105 | 0.7663
0.8337 | **0.0898**
**0.1411** | 81.03
74.68 |
| **Class 3** | **Conv.-Diff.-Reac.** | 1~5 | Abs. err.
Rel. err. | 0.1663
0.2057 | 0.0865
0.1415 | 0.4943
0.6104 | **0.0311**
**0.0525** | 64.02
62.88 |
| | | 1~10 | Abs. err.
Rel. err. | 0.5321
0.5928 | 0.3170
0.3772 | 0.7051
0.8027 | **0.0508**
**0.0939** | 83.98
75.10 |
| | | 1~20 | Abs. err.
Rel. err. | 0.7450
0.7960 | 0.4080
0.4645 | 0.7136
0.8100 | **0.0353**
**0.0812** | 91.94
82.88 |

# I FINE-TUNING P$^2$INNS

In general, our P$^2$INNs outperform other baselines in most of the tested equations. We can fine-tune the pre-trained model to further increase the accuracy and in this section, we show the efficacy of the fine-tuning step with intuitive visualizations.

## I.1 EXPERIMENTS WITH GAUSSIAN DISTRIBUTION AS AN INITIAL CONDITION

Experiments summarized in Table 3 use the initial condition of the Gaussian distribution $N(\pi, (\pi/2)^2)$. We fine-tune P$^2$INN from Table 3 on two PDEs: a convection equation with $\beta = 10$, and a reaction equation with $\rho = 5$. For the coefficient range used in pre-training, we select $\beta \in [1, 20]$ and $\rho \in [1, 10]$, respectively. We compare our fine-tuned model with vanilla PINN and results are summarized in Figure 8.

For the additional study, we show how the results of pre-trained P$^2$INNs are affected by varying the PDE parameters. Figures 9(a-c)/(d-f) are the results of convection/reaction equations. As shown in Figure 9, our P$^2$INNs effectively learn the differences among the various coefficient settings.

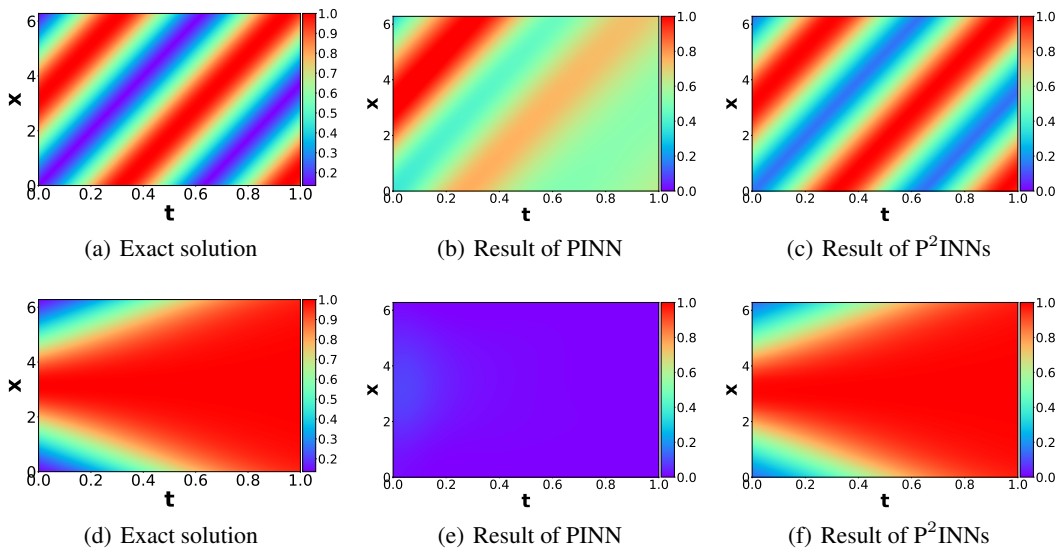

Figure 8: Experimental results of fine-tuning P$^2$INN. Convection equation of $\beta = 30$ (Figure 8. (a)-(c)). Reaction equation of $\rho = 5$ (Figure 8. (d)-(f)). Figures 8 (c) and (f) are the results after fine-tuning, and the results before fine-tuning can be checked in Figure 9.

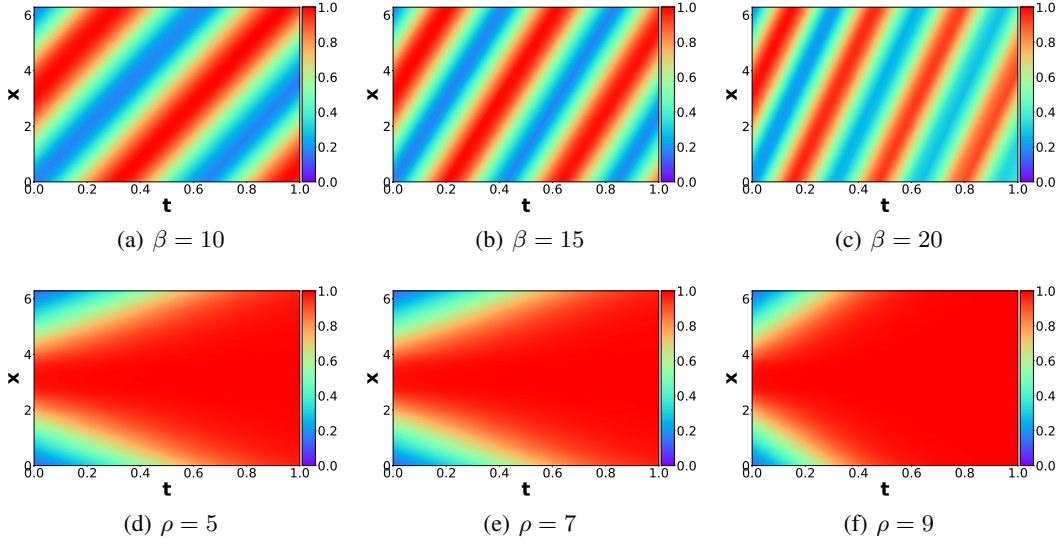

Figure 9: Results of P$^2$INNs on convection equation and reaction equation without fine-tuning.

## I.2 FAILURE MODE

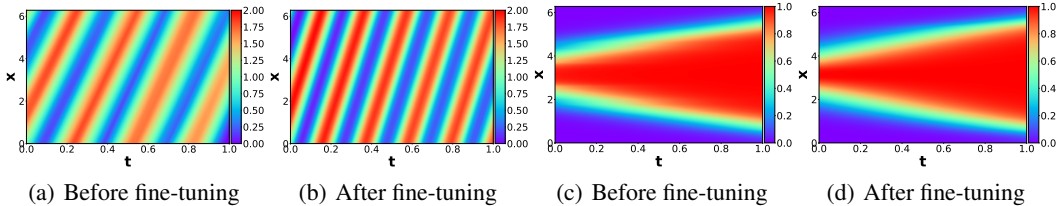

|                 (a) Before fine-tuning |                 (b) After fine-tuning |                 (c) Before fine-tuning |                 (d) After fine-tuning |

Figure 10: Experimental results of P$^2$INN in Section 3. Figures (a) and (b) are the results of convection equation $\beta = 30$, and Figures (c) and (d) are reaction equation $\rho = 5$.

Table 4: Results of P$^2$INNs for the failure mode. We use a convection equation with $1 + \sin(x)$ as an initial condition and a reaction equation with the Gaussian distribution $N(\pi, (\pi/4)^2)$.

| Failure mode | PINN | | P$^2$INN | |
|---|---|---|---|---|
| | Abs. err. | Rel. err. | Abs. err. | Rel. err. |
| $\beta = 30$ | 0.6132 | 0.5734 | 0.0910 | 0.0916 |
| $\rho = 5$ | 0.5490 | 0.9844 | 0.0058 | 0.0173 |

Figure 4 is the result of P$^2$INN for the failure mode, and Figure 10 is a comparison between before and after fine-tuning on the results of P$^2$INN. Figures 10 (a) and (b) are the results on convection equation of $\beta = 30$, and Figures 10 (c) and (d) are the results on reaction equation of $\rho = 5$. As shown in Table 4, P$^2$INN significantly improves the performance compared to PINN.

## J EXPERIMENTAL RESULTS ON 2D HELMHOLTZ EQUATION

We undertake an evaluation by training our P$^2$INN model on a 2D Helmholtz equation and subsequently comparing its performance with that of PINNs. In the case of $a = \{2.50, 2.70, 2.80, 3.00\}$, performance is evaluated on the seen PDEs utilized for training, while for $a = \{2.65, 2.75, 2.85\}$, performance is assessed on the unseen PDEs not used during training phase. All test datasets consist of data that is not employed in the training, and the experimental results are reported in Table 5 and Figure 11.

Table 5: Comparision with PINN, PINN-R and P$^2$INN on 2D Helmholtz equations

| Model | Metrics | $a = 2.50$ | $a = 2.65$ | $a = 2.70$ | $a = 2.75$ | $a = 2.80$ | $a = 2.85$ | $a = 3.00$ |
|---|---|---|---|---|---|---|---|---|
| **PINN** | Abs. err. | 0.1484 | 0.9077 | 1.9105 | 1.8942 | 1.5689 | 0.9077 | 2.4981 |
| | Rel. err. | 0.4817 | 2.0937 | 4.9264 | 4.7584 | 3.3739 | 2.0937 | 6.1532 |
| **PINN-R** | Abs. err. | 0.1107 | 0.2916 | 1.1590 | 1.4000 | 1.1095 | 1.5789 | 1.8800 |
| | Rel. err. | 0.3830 | 0.7239 | 2.8633 | 3.6641 | 2.6792 | 3.8059 | 4.7755 |
| **P$^2$INN** | Abs. err. | **0.0240** | **0.0259** | **0.0257** | **0.0263** | **0.0321** | **0.0232** | **0.0315** |
| | Rel. err. | **0.0718** | **0.0767** | **0.0788** | **0.0840** | **0.0975** | **0.0642** | **0.0973** |

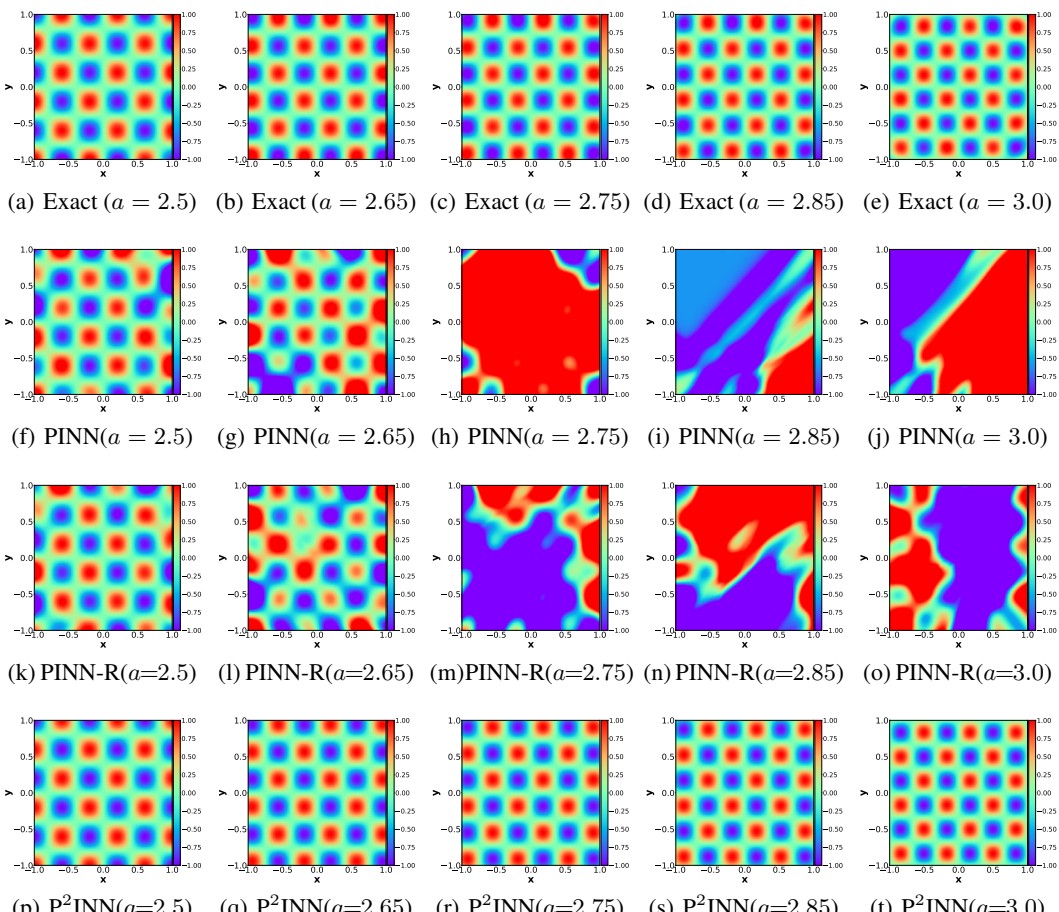

(a) Exact ($a = 2.5$)  (b) Exact ($a = 2.65$)  (c) Exact ($a = 2.75$)  (d) Exact ($a = 2.85$)  (e) Exact ($a = 3.0$)

(f) PINN($a = 2.5$)  (g) PINN($a = 2.65$)  (h) PINN($a = 2.75$)  (i) PINN($a = 2.85$)  (j) PINN($a = 3.0$)

(k) PINN-R($a$=2.5)  (l) PINN-R($a$=2.65)  (m) PINN-R($a$=2.75)  (n) PINN-R($a$=2.85)  (o) PINN-R($a$=3.0)

(p) P$^2$INN($a$=2.5)  (q) P$^2$INN($a$=2.65)  (r) P$^2$INN($a$=2.75)  (s) P$^2$INN($a$=2.85)  (t) P$^2$INN($a$=3.0)

Figure 11: [2D-Helmholtz equation] Exact solutions and results of PINN, PINN-R and P$^2$INN for various $a$

Table 6: Number of model parameters.

|         | PINN   | PINN-R | PINN-seq2seq | LargePINN | PINN-P | $P^2$INN |
|---------|--------|--------|--------------|-----------|--------|----------|
| #params | 10,401 | 10,401 | 10,401       | 82,941    | 91,651 | 76,851   |

Table 7: The relative and absolute $L_2$ errors over all the equations. Our $P^2$INNs surpass LargePINN and PINN-P in all but one cases, even without fine-tuning.

|         | PDE type         | Metric    | PINN   | LargePINN | PINN-P     | P2INN      |
|---------|------------------|-----------|--------|-----------|------------|------------|
| Class 1 | Convection       | Abs. err. | 0.1140 | 0.1191    | 0.0209     | **0.0198** |
|         |                  | Rel. err. | 0.1978 | 0.2084    | **0.0410** | 0.0464     |
|         | Diffusion        | Abs. err. | 0.6782 | 0.5868    | 0.3800     | **0.1916** |
|         |                  | Rel. err. | 1.2825 | 1.0994    | 0.7912     | **0.3745** |
|         | Reaction         | Abs. err. | 0.7902 | 0.7910    | 0.8975     | **0.0042** |
|         |                  | Rel. err. | 0.8460 | 0.8469    | 0.9908     | **0.0092** |
| Class 2 | Conv.-Diff.      | Abs. err. | 0.2735 | 0.1626    | 0.1253     | **0.0622** |
|         |                  | Rel. err. | 0.5106 | 0.3189    | 0.3009     | **0.1495** |
|         | Reac.-Diff.      | Abs. err. | 0.7167 | 0.7399    | 0.1756     | **0.0898** |
|         |                  | Rel. err. | 0.7998 | 0.8186    | 0.2632     | **0.1411** |
| Class 3 | Conv.-Diff.-Reac.| Abs. err. | 0.7450 | 0.7415    | 0.8590     | **0.0353** |
|         |                  | Rel. err. | 0.7960 | 0.7915    | 0.9532     | **0.0812** |

## K  ABLATION STUDIES ON PINN-P AND LARGEPINN

For more comprehensive evaluation, we conduct additional ablation studies following the experimental settings of Table 3 with the coefficient range of $1 \sim 20$ using PINN-P (cf. Appendix D.2) and LargePINN, which is PINN with bigger network size. As shown in Table 6, since the model size of our proposed $P^2$INN is larger than original PINN, we conduct experiments using a LargePINN model. The LargePINN has the same MLP architecture as the original PINN but with increased hidden dimensions, resulting in a model size of 82,941.

The experimental results of LargePINN, PINN-P, and $P^2$INN are summarized in Table 7. In all scenarios, as indicated by Table 7, the LargePINN model consistently performs inferiorly compared to $P^2$INNs, and $P^2$INNs outperforms PINN-P in all cases except one. That is, while the baselines struggles when learning the equations encompassing wide coefficient ranges, i.e., $1 \sim 20$. For instance, considering Conv.-Reac.-Diff. equation, the $L_2$ absolute error exhibited by $P^2$INN is 0.0353 whereas LargePINN and PINN-P have errors of 0.7415 and 0.8590, respectively. Note that this collective outcome underscores that $P^2$INN's separation of PDE parameters and spatial/temporal coordinates during the learning process significantly enhances both generalization capabilities and scalability.

