# OpenReview forum: "Extension of Physics-informed Neural Networks to Solving Parameterized PDEs"
_ICLR.cc/2024/Workshop/AI4DiffEqtnsInSci — AI4DiffEqtnsInSci @ ICLR 2024 Poster_

### Official Review · Reviewer_3J19 · 2024-02-28
**A notable work based on the reduction of the errors with regards to some baselines and the application of physics knowledge in machine learning.**

**Rating:** 7
**Confidence:** 3

**Review:**

The level of detail of the methods implemented in this work is excellent. It is relatively straightforward to understand what the authors did. Also, they state very clearly the motivation and the novelty of their study, and it helps to capture the importance of their contribution. I consider that this work is worth accepting it.

Pros:
* The topic of their work (physics-informed machine learning models) is very relevant in the scientific community currently.
* They provide all relevant information of the data, model architecture, baselines, training procedure and objective. The paper is very informative.
* The results reached are quite impressive, specially for the reaction equations, with regards to the tested baselines.
* With their proposed model they are tackling two weaknesses of the CDF equations, so it is more efficient that all other methods that tackle each weakness separately.

Cons:
* I appreciate that the authors explained the meaning of the equations and most of their parameters. However, since this is a Physics-informed machine learning model, I feel they should have explained also the physical meaning of the tested values of the parameters. In most of the cases they only say which numbers they used, but not actual physical explanation of most of them.
* The bibliography is quite old. There is not any mention of related work from 2023 nor 2024, (the most updated one is only one work from 2022), which makes me think that they are not comparing their method to a more state-of-the-art model.

---

### Meta-Review · Area_Chair_hxxA · 2024-02-28

**Recommendation:** Accept (Poster)

**Metareview:**

The reviewer mentions the novelty, clarity and relevance of the proposed work to the workshop. The authors tackle the problems with PINNs and verify their proposed approach on the failure modes paper from Krishnapriyan et al., 2021. I also vote for acceptance and would like the authors to address the reviewer's comments on the meaning of the physical PDE parameters and updated references in the final version. Also, the test problems such as convection are cited in the appendix from Krishnapriyan et al., 2021, which would also be good to add to the main body.

---

### Decision · Program_Chairs · 2024-02-28

Accept (Poster)